# Error Detection for Text-to-SQL Semantic Parsing

**Shijie Chen   Ziru Chen   Huan Sun   Yu Su**
The Ohio State University
{chen.10216, chen.8336, sun.397, su.809}@osu.edu

## Abstract

Despite remarkable progress in text-to-SQL semantic parsing in recent years, the performance of existing parsers is still far from perfect. Specifically, modern text-to-SQL parsers based on deep learning are often over-confident, thus casting doubt on their trustworthiness when deployed for real use. In this paper, we propose a parser-independent error detection model for text-to-SQL semantic parsing. Using a language model of code as its bedrock, we enhance our error detection model with graph neural networks that learn structural features of both natural language questions and SQL queries. We train our model on realistic parsing errors collected from a cross-domain setting, which leads to stronger generalization ability. Experiments with three strong text-to-SQL parsers featuring different decoding mechanisms show that our approach outperforms parser-dependent uncertainty metrics. Our model could also effectively improve the performance and usability of text-to-SQL semantic parsers regardless of their architectures[1].

## 1 Introduction

Recent years have witnessed a renewed interest in text-to-SQL semantic parsing (Bogin et al., 2019; Lin et al., 2020; Wang et al., 2020; Rubin and Berant, 2021; Cao et al., 2021; Gan et al., 2021; Scholak et al., 2021; Qi et al., 2022; Li et al., 2023), which allows users with a limited technical background to access databases through a natural language interface. Although state-of-the-art semantic parsers have achieved remarkable performance on Spider (Yu et al., 2018), a large-scale cross-domain text-to-SQL benchmark, their performance is still far from satisfactory for real use. While syntax errors can be automatically caught by SQL execution engines, detecting semantic errors in executable SQL queries can be non-trivial and time-consuming

even for experts (Jorgensen and Shepperd, 2007; Weiss et al., 2007). Therefore, an accurate error detector that can flag parsing issues and accordingly trigger error correction procedures (Chen et al., 2023) can contribute to building better natural language interfaces to databases.

Researchers have proposed multiple approaches for error detection in text-to-SQL parsing. Yao et al. (2019, 2020) detect errors by setting a threshold on the prediction probability or dropout-based uncertainty of the base parser. However, using these parser-dependent metrics requires the base parser to be calibrated, which limits their applicability. Several interactive text-to-SQL systems detect parsing errors based on uncertain span detection (Gur et al., 2018; Li et al., 2020; Zeng et al., 2020). Despite having high coverage for errors, this approach is reported to be of low precision. Finally, text-to-SQL re-rankers (Yin and Neubig, 2019; Kelkar et al., 2020; Bogin et al., 2019; Arcadinho et al., 2022), which estimate the plausibility of SQL predictions, can be seen as on-the-fly error detectors. Nevertheless, existing re-rankers are trained on in-domain parsing errors, limiting their generalization ability.

In this work, we propose a generalizable and parser-independent error detection model for text-to-SQL semantic parsing. Since syntax errors can be easily detected by an execution engine, we focus on detecting semantic errors in executable SQL predictions. We start developing our model with CodeBERT (Feng et al., 2020), a language model pre-trained on multiple programming languages. On top of that, we use graph neural networks to capture compositional structures in natural language questions and SQL queries to improve the performance and generalizability of our model. We train our model on parsing mistakes collected from a realistic cross-domain setting, which is indispensable to the model's strong generalization ability. Furthermore, we show that our model is versatile and can be used for multiple tasks, including er-

---

[1]Our implementation is available at https://github.com/OSU-NLP-Group/Text2SQL-Error-Detection.

ror detection, re-ranking, and interaction triggering. To summarize, our contributions include:

- We propose the first generalizable and parser-independent error detection model for text-to-SQL parsing that is effective on multiple tasks and different parser designs without any task-specific adaptation. Our evaluations show that the proposed error detection model outperforms parser-dependent uncertainty metrics and could maintain its high performance under cross-parser evaluation settings.

- Our work is the first comprehensive study on error detection for text-to-SQL parsing. We evaluate the performance of error detection methods on both correct and incorrect SQL predictions. In addition, we show through simulated interactions that a more accurate error detector could significantly improve the efficiency and usefulness of interactive text-to-SQL parsing systems.

## 2 Related Work

### 2.1 Text-to-SQL Semantic Parsing

Most existing neural text-to-SQL parsers adopt three different decoding mechanisms. The first one is sequence-to-sequence with constrained decoding, where a parser models query synthesis as a sequence generation task and prunes syntactically invalid parses during beam search. Several strong text-to-SQL parsers apply this simple idea, including BRIDGE v2 (Lin et al., 2020), PICARD (Scholak et al., 2021), and RESDSQL (Li et al., 2023). Another popular decoding mechanism is grammar-based decoding (Yin and Neubig, 2017), where parsers first synthesize an abstract syntax tree based on a pre-defined grammar and then convert it into a SQL query. Parsers using intermediate representations, such as IR-Net (Guo et al., 2019) and NatSQL (Gan et al., 2021) also fall into this category. Grammar-based decoding ensures syntactic correctness but makes the task harder to learn due to the introduction of non-terminal syntax tree nodes. Different from the above autoregressive decoding strategies, SmBoP (Rubin and Berant, 2021) applies bottom-up decoding where a SQL query is synthesized by combining parse trees of different depths using a ranking module. We evaluate our model with semantic parsers using each of these three decoding strategies and show that our model is effective on all of them.

### 2.2 Re-ranking for Text-to-SQL Parsing

Noticing the sizable gap between the accuracy and beam hit rate of semantic parsers, researchers have explored building re-ranking models to bridge this gap and improve parser performance. Global-GNN (Bogin et al., 2019) re-ranks beam predictions based on the database constants that appear in the predicted SQL query. This re-ranker is trained together with its base parser. More recently, Bertrand-DR (Kelkar et al., 2020) and T5QL (Arcadinho et al., 2022) fine-tune a pre-trained language model for re-ranking. However, both report directly re-ranking all beams using re-ranker scores hurts performance. To get performance gain from re-ranking, Bertrand-DR only raises the rank of a prediction if its re-ranking score is higher than the preceding one by a threshold. T5QL combines re-ranking score and prediction score by a weighted sum. Both approaches require tuning hyper-parameters. In contrast, when directly using the proposed parser-independent error detection model as re-rankers, we observe performance improvement on BRIDGE v2 and NatSQL without any constraint, showing that our approach is more generalizable and robust.

### 2.3 Interactive Text-to-SQL Parsing Systems

Interactive text-to-SQL parsing systems improve the usability of text-to-SQL semantic parsers by correcting potential errors in the initial SQL prediction through interactive user feedback. MISP (Yao et al., 2019, 2020) initiates interactions by setting a confidence threshold for the base parser's prediction probability. While this approach is intuitive, it requires the base parser to be well-calibrated when decoding, which does not hold for most modern parsers using deep neural networks. In addition, this design can hardly accommodate some recent parsers, such as SmBoP (Rubin and Berant, 2021), whose bottom-up decoding mechanism does not model the distribution over the output space. Several other interactive frameworks (Gur et al., 2018; Li et al., 2020; Zeng et al., 2020) trigger interactions when an incorrect or uncertain span is detected in the input question or predicted SQL query. While these approaches have high coverage for parsing errors, they tend to trigger unnecessary interactions for correct initial predictions. For example, PIIA (Li et al., 2020) triggers interactions on 98% of the questions on Spider's development set when its base parser has an accuracy of 49%.

| Parser | Train | | | Development | | | Test | | |
|---|---|---|---|---|---|---|---|---|---|
| | #Beam | Hit | Miss | #Beam | Hit | Miss | #Beam | Hit | Miss |
| SmBoP | 5322 | 6062/1.4 | 12937/2.6 | 1416 | 1864/1.3 | 3159/2.5 | 989 | 1324/1.3 | 1498/1.5 |
| BRIDGE v2 | 5482 | 3898/0.7 | 8932/1.6 | 1465 | 1116/0.8 | 2057/1.4 | 1031 | 852/0.8 | 1096/1.1 |
| NatSQL | 5398 | 7095/1.3 | 13443/2.5 | 1474 | 2207/1.5 | 3522/2.4 | 1030 | 1582/1.5 | 2584/2.5 |

Table 1: Statistics of the data collected from three base parsers. #Beam: number of beams that have executable predictions. For beam hit and misses, we report total/average_per_beam.

Compared to these methods, the proposed method strikes a better balance between performance and efficiency, and thus could improve the user experience of interactive text-to-SQL parsing systems.

## 3 Parser-independent Error Detection

### 3.1 Problem Formulation

Given a question $X = \{x_1, x_2, \cdots, x_m\}$ and a SQL query $\hat{y} = \{\hat{y}_1, \hat{y}_2, \cdots, \hat{y}_n\}$ predicted by a text-to-SQL parser, the error detection model estimates the probability of $\hat{y}$ being correct, denoted by $s$:

$$s = p(\hat{y} = y^*|X, \hat{y})$$

We perform error detection and action triggering by setting a threshold for $s$. For re-ranking, we directly use $s$ as the ranking score without modification.

### 3.2 Cross-domain Error Collection

We consider two factors that could lead to text-to-SQL parsing errors: *insufficient training data* and the *cross-domain generalization gap*. To simulate such errors, we collect data from weak versions of base parsers in a cross-domain setting. More specifically, we split the Spider training set into two equal-sized subsets by databases and train the base parser on each subset. Then we perform inference on the complementary subset and collect beam predictions as data for error detection. We keep executable SQL queries and label them based on execution accuracy. We use a fixed version of Spider's official evaluation script (Appendix A) and keep up to five parser predictions for each question after deduplication. The collected samples are divided into training and development sets by an 80:20 ratio according to databases as well. In this way, we get high-quality training data for our error detection model in a setting that approximates the real cross-domain testing environment. For testing, we train each base parser on the full Spider training set and collect executable beam predictions on the Spider development set. Beams with un-executable top predictions are skipped. We report the number of beams, total number of question-SQL pairs, and average number of such pairs per beam for each split in Table 1. Following existing literature (Kelkar et al., 2020), we refer to correct SQL queries in the beam as beam hits, and incorrect ones as beam misses.

We notice that BRIDGE v2 generates significantly fewer executable SQL queries on all data splits. This is due to its unconstrained decoder with rule-based filtering. In addition to that, BRIDGE v2 generates a default SQL query that counts the number of entries in the first table when there are no valid predictions in the beam. Although this mechanism ensures the parser generates at least one executable query, such negative samples do not fit the overall error distribution and may harm the error detection model. NatSQL and SmBoP take into account grammatical constraints of SQL during decoding and thus could generate more executable queries than BRIDGE v2. Table 1 also shows that NatSQL and SmBoP produce a similar amount of beam hits and beam misses on the training and development splits. However, the number of executable beam misses generated by SmBoP on the test split is noticeably lower, while the behavior of NatSQL is more consistent.

### 3.3 Model Architecture

Figure 1 illustrates the architecture of the proposed error detection models. We use Code-BERT (Feng et al., 2020) as our base encoder to jointly encode the input question and SQL query. Following CodeBERT's input construction during pre-training, we concatenate questions and SQL queries with special tokens, namely $[CLS], x_1, \cdots, x_m, [SEP], \hat{y}_1, \cdots, \hat{y}_n, [EOS]$ as input and obtain their contextualized representations $h_X$ and $h_{\hat{y}}$. We only use question and SQL as input since we found in preliminary experiments that adding database schema information (table and column names) in the input hurts performance.

In light of the compositional nature of questions

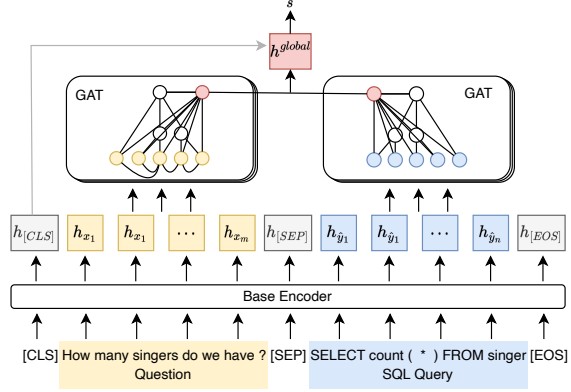

Figure 1: Architecture of our error detection models.

and SQL queries, we propose to model their structural features via graph neural networks. For natural language questions, we obtain their dependency parse trees and constituency parse trees from Stanza (Qi et al., 2020) and merge them together. This is possible since edges in dependency parse trees are between two actual tokens, which corresponds to leaf nodes in constituency parse trees. For SQL queries, we extract their abstract syntax trees via Antlr4.[2] To make the input graphs more compact and lower the risk of overfitting, we further simplify the parse trees by removing non-terminal nodes that only have one child in a top-down order. Additionally, for SQL queries, we remove the subtrees for join constraints which do not carry much semantic information but are often quite long. At last, we add sequential edges connecting the leaf nodes in the parse trees by their order in the original questions and SQL queries to preserve natural ordering features during graph learning.

We initialize the representations of parse tree leaf nodes with CodeBERT's contextualized representations and randomly initialize representations of other nodes according to their types in the parse tree. The two input graphs are encoded by two separate 3-layer graph attention networks (Brody et al., 2022). Then we obtain the global representation of each graph via average pooling and concatenate them to get an aggregated global representation:

$$h^{global} = [h_X^{global}; h_{\hat{y}}^{global}]$$

We denote models with graph encoders as Code-BERT+GAT in Section 4. When simply fine-tuning CodeBERT, $h^{global} = h_{[CLS]}$.

[2]Antlr: https://www.antlr.org/, more details in Appendix D.

Finally, a 2-layer feed-forward neural network with tanh activation is used to score the aggregated representation $v$. The score $s$ for each input question-SQL pair is:

$$s = p(\hat{y} = y^*|X, \hat{y}) = \sigma(\text{FFN}(h^{global}))$$

where $y^*$ is the gold SQL query and $\sigma$ represents the sigmoid function. We train our model by minimizing a binary cross entropy loss:

$$\mathcal{L} = \mathbb{1}_{\hat{y}=y^*} \cdot \log s + (1 - \mathbb{1}_{\hat{y}=y^*}) \cdot \log(1 - s)$$

During training, we supply the model with samples from $K$ beams at each step, where $K$ is the batch size.

## 4 Experiments

In this section, we first evaluate the performance (Section 4.2.1) and generalization ability (Section 4.2.2) of our error detection model on the binary error detection task. Then we investigate our model's effectiveness when used for re-ranking (Section 4.2.3) and action triggering (Section 4.2.4).

### 4.1 Experiment Setup

**Baseline Methods** We compare our parser-independent error detectors with parser-dependent uncertainty metrics, including prediction probability and dropout-based uncertainty. Since SmBoP (Rubin and Berant, 2021) uses bottom-up decoding which separately scores and ranks each candidate prediction, we deduplicate SmBoP's beam predictions by keeping the maximum score and perform softmax on the deduplicated beam to get a probability distribution over candidate predictions, which can be seen as a reasonable approximation to its confidence. BRIDGE v2 (Lin et al., 2020) and NatSQL (Gan et al., 2021) use autoregressive decoders, and we directly use the log probability of its prediction as its confidence score. Probability-based methods are denoted by superscript $p$. In terms of dropout-based uncertainty, we follow MISP (Yao et al., 2019) and measure the standard deviation of the scores (SmBoP) or log probability (BRIDGE v2 and NatSQL) of the top-ranked prediction in 10 passes. Dropout-based uncertainty is denoted by superscript $s$.

**Evaluation Metrics** We first evaluate our model on the error detection task. After that, we test performance when it is used for re-ranking and action triggering.

| Parser | Model | Positive | | | Negative | | | Acc | AUC |
|--------|-------|----------|------|------|----------|------|------|------|------|
| | | Precision | Recall | F1 | Precision | Recall | F1 | | |
| SmBoP | SmBoP$^p$ | 80.8 | **94.5** | 86.6 | 52.1 | 18.9 | 23.9 | 77.4 | 67.0 |
| | SmBoP$^s$ | 81.5 | 91.9 | 85.7 | 56.6 | 25.3 | 29.4 | 76.9 | 79.2 |
| | CodeBERT | 82.9 | 92.6 | 86.7 | **60.8** | 33.9 | 36.3 | 78.3 | 80.8 |
| | CodeBERT+GAT | **85.0** | 90.6 | **87.2** | 56.7 | **44.4** | **46.4** | **79.8** | **81.7** |
| BRIDGE v2 | BRIDGE v2$^p$ | 77.9 | 85.1 | 80.9 | 60.9 | 50.4 | 54.0 | 73.2 | 77.8 |
| | BRIDGE v2$^s$ | 76.1 | 83.6 | 79.5 | 57.9 | 46.0 | 50.8 | 71.3 | 77.7 |
| | CodeBERT | 78.6 | **87.6** | 82.4 | 69.1 | 51.1 | 57.0 | 75.3 | 81.3 |
| | CodeBERT+GAT | **80.9** | 86.4 | **83.4** | **69.5** | **58.7** | **63.2** | **77.6** | **83.0** |
| NatSQL | NatSQL$^p$ | 78.1 | **93.2** | 84.6 | 67.3 | 36.1 | 45.4 | 76.3 | 79.2 |
| | NatSQL$^s$ | 77.0 | 91.4 | 83.0 | 62.8 | 33.1 | 40.3 | 74.0 | 76.2 |
| | CodeBERT | 84.6 | 90.8 | **87.3** | **72.3** | 60.5 | 64.6 | **81.8** | 86.5 |
| | CodeBERT+GAT | **86.6** | 87.4 | 86.8 | 68.5 | **68.1** | **67.0** | 81.7 | **86.9** |

Table 2: Error detection performance with three base parsers on Spider's development set. We highlight the best performance with each parser in bold.

For error detection, we report precision, recall, and F1 scores for each method on both positive and negative samples. However, these metrics depend on the threshold used. To more comprehensively evaluate the overall discriminative ability of each method, we present the area under the receiver operating characteristic curve (AUC), which is not affected by the choice of threshold. We apply 5-fold cross-validation and report performance using the threshold that maximizes the accuracy of each method. Test samples are partitioned by databases.

For the re-ranking task, we evaluate on the final beam predictions of fully trained base parsers on Spider's development set and report top-1 accuracy.

For action triggering, we evaluate system performance under two settings: answer triggering and interaction triggering. In answer triggering, we measure system answer precision when answering different numbers of questions. In interaction triggering, we measure system accuracy using different numbers of interactions.

Error detection and re-ranking results are average performance over 3 different random seeds. For action triggering, we evaluate checkpoints with the highest accuracy on the development split of our collected data.

**Implementation** Our models are trained with a batch size of 16 and are optimized by the AdamW (Loshchilov and Hutter, 2019) optimizer with default parameters. Training lasts 20 epochs with a learning rate of 3e-5 following a linear decay schedule with 10% warm-up steps. All models are trained on an NVIDIA RTX A6000 GPU.

## 4.2 Results

### 4.2.1 Error Detection

To evaluate error detection methods in a realistic setting, we use final SQL predictions made by SmBoP, BRIDGE v2, and NatSQL on Spider's development set that are executable as test datasets. As shown in Table 2, the dropout-based uncertainty measure significantly outperforms the approximate confidence measure on negative samples (29.4 vs 23.9 in negative F1) with SmBoP. However, we notice the opposite with BRIDGE v2 and NatSQL, which is consistent with the observation of MISP (Yao et al., 2019) that is based on a parser with autoregressive decoder as well. Nonetheless, the dropout-based uncertainty measure is still indicative of these two parsers. We also notice that parser-dependent metrics exhibit high recall and low precision on positive samples, showing that the three parsers, despite using different decoding strategies, are over-confident in their predictions.

On all three parsers, our proposed error detector significantly outperforms the two parser-dependent uncertainty metrics, especially on negative samples. With the added structural features, CodeBERT+GAT further improves overall error detection performance, especially in recall on incorrect predictions (8.6% absolute improvement on average), which indicates structural features could help the model learn more generalizable error patterns. We also find that the advantage of CodeBERT+GAT mainly comes from its higher performance on *hard* and *extra hard* questions (Appendix C). Compared to parser-dependent metrics, our

| Target | Source | Model | Positive | | | Negative | | | Acc | AUC |
|--------|--------|-------|-----------|--------|------|-----------|--------|------|------|------|
| | | | Precision | Recall | F1 | Precision | Recall | F1 | | |
| SmBoP | - | SmBoP$^p$ | 80.8 | 94.5 | 86.6 | 52.1 | 18.9 | 23.9 | 77.4 | 67.0 |
| | | SmBoP$^s$ | 81.5 | 91.9 | 85.7 | 56.6 | 25.3 | 29.4 | 76.9 | **79.2** |
| | BRIDGE v2 | CodeBERT | 80.3 | 94.8 | 86.4 | 52.9 | 16.3 | 20.3 | 77.2 | 75.9 |
| | | CodeBERT+GAT | 80.0 | **95.9** | 86.8 | 44.1 | 14.4 | 18.8 | 77.8 | 77.0 |
| | NatSQL | CodeBERT | 81.9 | 95.2 | **87.7** | **58.2** | 23.4 | 30.8 | 79.6 | 75.9 |
| | | CodeBERT+GAT | **83.1** | 93.3 | 87.6 | 56.0 | **31.5** | **38.1** | **79.9** | 78.2 |
| BRIDGE v2 | - | BRIDGE v2$^p$ | 77.9 | 85.1 | 80.9 | 60.9 | 50.4 | 54.0 | 73.2 | 77.8 |
| | | BRIDGE v2$^s$ | 76.1 | 83.6 | 79.5 | 57.9 | 46.0 | 50.8 | 71.3 | 77.7 |
| | SmBoP | CodeBERT | 78.4 | **88.2** | 82.4 | 70.1 | 50.1 | 55.6 | 75.0 | 82.4 |
| | | CodeBERT+GAT | **80.2** | 87.9 | **83.5** | **71.5** | **55.5** | **61.1** | **77.1** | **83.1** |
| | NatSQL | CodeBERT | 78.5 | **88.2** | 82.9 | 68.6 | 50.1 | 56.9 | 75.9 | 80.0 |
| | | CodeBERT+GAT | 78.9 | 87.2 | 82.5 | 68.2 | 51.7 | 56.9 | 75.5 | 81.7 |
| NatSQL | - | NatSQL$^p$ | 78.1 | **93.2** | 84.6 | **67.3** | 36.1 | 45.4 | 76.3 | 79.2 |
| | | NatSQL$^s$ | 77.0 | 91.4 | 83.0 | 62.8 | 33.1 | 40.3 | 74.0 | 76.2 |
| | SmBoP | CodeBERT | **83.7** | 86.1 | 84.0 | 63.5 | **61.1** | 58.5 | 77.0 | 85.2 |
| | | CodeBERT+GAT | 83.5 | 87.2 | 84.7 | 65.1 | 59.5 | **59.9** | **78.2** | **85.7** |
| | BRIDGE v2 | CodeBERT | 79.7 | 89.7 | 84.2 | 65.3 | 43.3 | 51.1 | 76.9 | 79.6 |
| | | CodeBERT+GAT | 81.1 | 89.2 | **84.8** | 65.8 | 48.0 | 54.7 | 78.1 | 81.5 |

Table 3: Cross-parser generalization performance with three base parsers on Spider's development set. We highlight the best performance with each target parser in bold.

model yields the largest performance gain in both accuracy and AUC with NatSQL and reasonable gains with the other two parsers, possibly due to the higher quality of its training data and better behavior consistency on the test split.

### 4.2.2 Cross-parser Generalization

We evaluate our models' cross-parser generalization ability by training error detectors on data collected from one parser and testing on the other two following the same 5-fold cross-validation setting. Table 3 summarizes cross-parser transfer performance on each parser. Even in this setting, our error detectors could still outperform parser-dependent metrics except for SmBoP, where our models fall short slightly in AUC.

On all parsers, we observe better performance on models trained with stronger parsers. For example, on SmBoP, the CodeBERT+GAT model trained with NatSQL is better than the one trained with BRIDGE v2 by 2.1% in accuracy and 1.2% in AUC. Meanwhile, the models trained with SmBoP perform the best on BRIDGE v2 and NatSQL in negative F1, accuracy, and AUC. We hypothesize errors made by stronger parsers are more diverse and of higher quality and thus allow models trained on them to generalize better to weaker parsers. We found that the errors generated by BRIDGE

v2 and NatSQL, two autoregressive parsers, are more likely to share prefixes and differ in simple operations, such as the choice of columns, aggregation functions, or logic operators (examples in Appendix E). In contrast, the bottom-up decoder of SmBoP generates more diverse errors with complex structures, such as subqueries and set operations. The higher diversity of SmBoP's predictions increases the coverage of the data collected from it, which contributes to the stronger generalization ability of the corresponding error detectors.

### 4.2.3 Re-ranking

| Re-ranker | SmBoP | BRIDGE v2 | NatSQL |
|-----------|-------|-----------|--------|
| N/A | **75.1** | 67.3 | 71.3 |
| RR | 72.2 | **69.7** | 72.5 |
| ED + RR | 73.7 | 68.9 | **73.9** |
| Beam Hit | 80.5 | 73.1 | 81.1 |

Table 4: Execution accuracy with re-ranking using the CodeBERT + GAT model. RR: Re-ranking all beams; ED+RR: Re-ranking beams after error detection.

We evaluate the re-ranking performance of our error detection models in two settings. In re-ranking-all (RR), we re-rank all beams based on the score assigned by the error detector. In error detection then re-ranking (ED+RR), we only re-rank

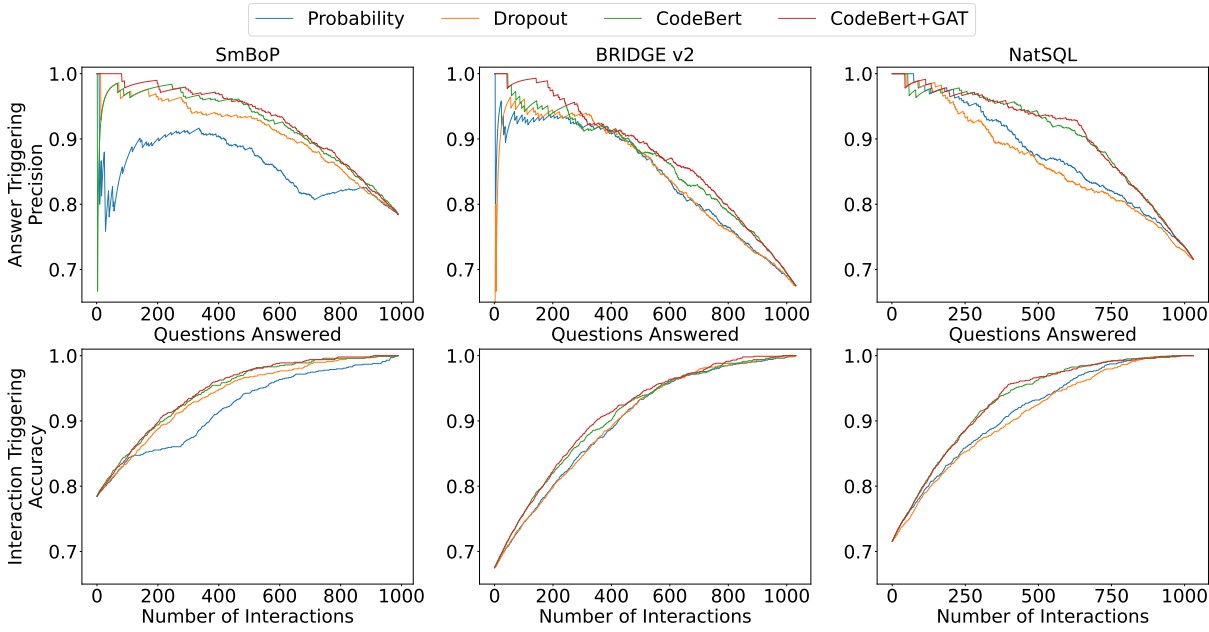

Figure 2: Performance in simulated interactive semantic parsing with three base parsers.

the beams whose top-ranked prediction has a score below a given threshold. For simplicity, we use a decision threshold of $0.5$ for error detection.

As shown in Table 4, our error detectors can improve the performance of BRIDGE v2 and NatSQL in both settings without training on any re-ranking supervision. Compared with existing re-rankers, our model does not need extra hyper-parameters for performance gain, even in the re-ranking-all setting. However, re-ranking hurts the performance of SmBoP. We attribute this to the larger train-test discrepancy due to the bottom-up nature of SmBoP's decoder. As discussed in Section 3.2 and Section 4.2.2, SmBoP produces more diverse beam predictions, but its behavior is less consistent on the test split. While the diversity benefits the quality of data for training error detectors, the inconsistency makes re-ranking on the test split harder. Although SmBoP is the strongest parser among the three, state-of-the-art text-to-SQL parsers predominantly use autoregressive decoders. Therefore, we still expect our approach to be generally applicable. We further perform 0-shot re-ranking evaluation on the more challenging KaggleDBQA (Lee et al., 2021) dataset (Appendix B). CodeBERT+GAT improves BRIDGE v2's accuracy from 20.5% to 21.8%, showing good generalization to unseen datasets.

#### 4.2.4 Action Triggering in Interactive Systems

In this section, we evaluate the potential gain of using our error detection model as an answer trig-

ger and interaction trigger in interactive semantic parsing systems.

**Answer triggering** When using error detectors for answer triggering, the interactive semantic parsing system restrain from answering the user's question when an error is detected. The upper half of Figure 2 demonstrates the change of precision when varying the decision threshold. In general, a high threshold $p$ (or lower $s$) reduces the number of questions answered for higher precision. Conversely, a lower $p$ (or higher $s$) encourages the system to answer more questions at the cost of making more mistakes.

Because of the high precision on positive samples, the proposed error detectors outperform both baseline methods and allow the system to answer more questions at higher precision. As shown by Table 5, when maintaining a precision of 95%, our error detectors allow the system to answer 76% to 175% more questions compared to parser-dependent metrics.

| Model | SmBoP | BRIDGE v2 | NatSQL |
|---|---|---|---|
| Probability | 8 | 24 | 302 |
| Dropout | 295 | 104 | 213 |
| CodeBERT | 498 | 247 | **441** |
| CodeBERT+GAT | **520** | **286** | 395 |

Table 5: The number of questions each parser could answer when maintaining a precision of 95%.

| Graph | Data Source | NatSQL | | | BRIDGE v2 | | SmBoP | |
|---|---|---|---|---|---|---|---|---|
| | | Acc | AUC | RR | Acc | AUC | Acc | AUC |
| NatSQL$^p$ | | 76.3 | 79.2 | 71.3 | - | - | - | - |
| Simplified | in-domain | 79.0 | 83.8 | 71.2 | 73.3 | 76.8 | 78.5 | 71.3 |
| Simplified | cross-domain | **81.7** | 86.9 | **72.5** | **75.5** | **81.7** | **79.9** | **78.2** |
| Original | cross-domain | **81.7** | **87.2** | 71.9 | 75.0 | 79.9 | 79.5 | 76.0 |

Table 6: Ablation results using the CodeBERT + GAT model trained on data collected from NatSQL. We report accuracy, AUC, re-ranking-all (RR) performance on NatSQL's test split as in-domain evaluation and report accuracy and AUC when tested on SmBoP and BRIDGE v2 as generalization evaluation. NatSQL$^p$ is included for reference.

| Model | SmBoP | BRIDGE v2 | NatSQL |
|---|---|---|---|
| Probability | 535 | 557 | 573 |
| Dropout | 412 | 566 | 597 |
| CodeBERT | 370 | 555 | 421 |
| CodeBERT+GAT | **356** | **538** | **384** |

Table 7: The number of interactions each parser needs with interaction triggering to reach an accuracy of 95%.

**Interaction triggering** We simulate the potential gain of more accurate interaction triggers by assuming oracle error correction interactions, where any detected error would be fixed through interactions with users. Ideally, we would want to get higher system accuracy with fewer interactions. The lower half of Figure 2 illustrates the change of accuracy at different interaction budgets.

Our parser-independent models consistently improve upon parser-dependent metrics, resulting in more efficient interactive semantic parsing systems. Due to higher precision and recall on erroneous base predictions, systems using our models could correct more errors and avoid unnecessary interactions. As shown by Table 7, depending on the base parser, our model brings a 3.4% to 33% reduction to the number of interactions required for reaching an accuracy of 95%.

### 4.3 Ablation

We perform ablation studies on the impact of cross-domain error collection and graph learning using the CodeBERT+GAT model. We report the models' accuracy, AUC, and re-ranking performance in the re-rank all setting (RR) on the test split of NatSQL. We also test the models on BRIDGE v2 and SmBoP to evaluate their generalization ability.

**Cross-domain error collection** We train a NatSQL model using the full Spider training set and perform inference on the same data set to get its beam predictions. Then we create training data for error detection following the procedure described in Section 3.2. In this way, we collect in-domain parsing errors in the same setting as Bertrand-DR and T5QL. As shown by Table 6, the error detector trained on in-domain errors significantly underperforms the one trained on cross-domain errors. The performance of NatSQL deteriorates after re-ranking, which is consistent with the findings of previous re-rankers. Thus, we conclude that collecting high-quality parsing errors in a realistic cross-domain setting is critical to building an accurate and generalizable error detector.

**Simplified graph input** In this setting, we do not simplify constituency parse trees and SQL abstract syntax trees when constructing input graphs for graph neural networks. Table 6 shows that the model without graph simplification slightly outperforms the one using simplified graphs in AUC. Despite that, its re-ranking and cross-parser generalization performance are lower. We hypothesize that graph simplification could maintain important structural features of the input and improve the model's generalization ability by alleviating overfitting during training.

## 5 Conclusion

In this work, we propose the first generalizable parser-independent error detection model for text-to-SQL semantic parsing. Through learning compositional structures in natural language and SQL queries, the proposed model significantly outperforms parser-dependent uncertainty metrics and could generalize well to unseen parsers. We further demonstrate the versatility of our approach in error detection, re-ranking, and action triggering through a case study with three state-of-the-art text-to-SQL parsers featuring different decoding mechanisms.

Our experiments highlight the important role of structural features and cross-domain training data in building strong and generalizable error detec-

tors for semantic parsing. Potential future work includes (1) developing more advanced architectures to better evaluate the semantic correctness of synthesized SQL queries, (2) exploring data synthesis strategies to automatically create high-quality training data for error detection models.

## Limitations

This work is the first attempt towards building a versatile error detector for text-to-SQL semantic parsing. Although our model is parser-independent, the current data collection process depends on the choice of base parsers. As a result, the collected data may inherit certain biases in the base parsers. Our experiments show that data collected from stronger base parsers helps the model to generalize to weaker parsers. However, how to collect high-quality training data for error detection with stronger base parsers like SmBoP remains an open problem. A promising future direction may be developing a comprehensive data synthesis approach to improve the quality of training data. Grappa (Yu et al., 2021) uses context-free grammar to synthesize SQL queries for pre-training Transformer encoders for text-to-SQL parsing. This approach could be adapted to generate syntactically correct but semantically incorrect SQL queries in a controllable way.

Another major limitation is that our current model does not consider database schema information. Since SQL queries are grounded in databases, in principle database schema (tables, columns, and foreign-key relationships) should be an important part of error detection. The common practice in text-to-SQL semantic parsing is to linearize the database schema and concatenate all table and column names to the input to the Transformer encoder. However, our preliminary experiments show that this operation actually hurts the error detection performance. A similar observation is also reported by Kelkar et al. (2020). Nevertheless, our approach performs strongly for error detection as it can still effectively capture semantic errors that are free from schema linking mistakes. This can be explained by the high column mention rate in Spider (Pi et al., 2022). Future work could develop more effective entity linking mechanisms to extend our model to more challenging testing environments where schema linking errors are more common.

## Acknowledgements

We would like to thank colleagues from the OSU NLP group for their thoughtful comments. This research was sponsored in part by a sponsored research award by Cisco Research, NSF IIS-1815674, NSF CAREER #1942980, NSF OAC-2112606, and Ohio Supercomputer Center (Center, 1987). The views and conclusions contained herein are those of the authors and should not be interpreted as representing the official policies, either expressed or implied, of the U.S. government. The U.S. Government is authorized to reproduce and distribute reprints for Government purposes notwithstanding any copyright notice herein.

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

## Appendices

We provide additional details as follows:

## A   Fixed Spider Evaluation Script

We fix the following problems in the official evaluation process of the Spider dataset.

1. Incorrect handling of UTF-8 encoded databases. We fix this issue in the Python implementation of the evaluation script.

2. Certain examples have gold queries that execute to an empty result, resulting in false positive labels. For these samples, we use exact set match score instead.

3. Considering the order of returned results when unnecessary. We fix this issue by sorting each column in the execution result in alphabetical order when there is no 'Order By' clause at the top level.

4. Duplicated values for queries with 'limit' clause. This results in false negatives for questions querying the maximum or minimum values of a column or aggregation. We fix this issue by removing the limit clause in the gold and predicted SQL when their argument is the same.

We show the accuracy of each parser on Spider's development set based on the official and our version of evaluation in Table A.1. Although the differences in accuracy are within 1%, the disagreements impact 3.3% to 4% of the examples. We use the fixed evaluation script to reduce false positive and false negative labels both for training data collection and evaluation of our model.

| Parser | Acc$^*$ | Acc | Disagreement |
|---|---|---|---|
| SmBoP | 75.0 | 75.3 | 41 |
| BRIDGE v2 | 67.3 | 68.3 | 34 |
| NatSQL | 71.3 | 71.9 | 40 |

Table A.1: Execution accuracy of the three parsers on Spider's development set using official evaluation script (Acc) and our fixed evaluation script (Acc$^*$). Disagreement counts the number of samples with different labels.

| Parser | Model | Easy (24.0%) | Medium (43.1%) | Hard (16.8%) | Extra Hard (16.1%) |
|---|---|---|---|---|---|
| SmBoP | CodeBERT | 85.0/84.9 | **81.0/80.5** | 75.6/81.8 | 63.3/76.4 |
| | CodeBERT+GAT | **87.1/86.9** | 80.5/78.8 | **80.6/82.7** | **65.1/76.6** |
| BRIDGE v2 | CodeBERT | **82.9**/81.7 | 74.6/76.8 | 75.7/85.7 | 64.4/75.1 |
| | CodeBERT+GAT | 82.7/**82.4** | **76.4/80.8** | **79.0/86.3** | **70.0/76.7** |
| NatSQL | CodeBERT | **89.0/91.9** | **82.6/84.6** | 78.5/84.6 | 72.1/78.5 |
| | CodeBERT+GAT | 88.7/91.6 | 81.4/84.0 | **78.9/87.3** | **74.1/80.1** |

Table A.2: Error detection performance Acc/AUC break down by difficulty on the Spider dev set. Difficulty is decided by the official Spider evaluation script. The proportion of each difficulty type is in the parenthesis.

## B    KaggleDBQA Results

| Re-ranker | CodeBERT+GAT |
|---|---|
| N/A | 20.5 |
| RR | **21.8** |
| ED + RR | 21.4 |
| Beam Hit | 25.9 |

Table B.3: Execution accuracy of BRIDGE v2 and re-ranking using the CodeBERT + GAT model (left) and CodeBERT (right) on KaggleDBQA. RR: Re-ranking all beams; ED+RR: Re-ranking beams after error detection.

To test the generalization abilities of the proposed error detecor, we perform 0-shot evaluations on the 370 test examples[3] in KaggleDBQA(Lee et al., 2021). KaggleDBQA features more realistic database naming and makes entity linking significantly more challenging than Spider. We only experiment with BRIDGE v2 for the following reasons: (1) Under the 0-shot testing, SmBoP trained on Spider got an accuracy of 1.6% both for top-1 and beam hit, making re-ranking meaningless. This is partly due to the failure of its entity linking modules based on span extraction from questions. (2) At the time of writing, the SQL-to-NatSQL part of NatSQL's preprocessing code has not been released and its current codebase does not support KaggleDBQA.

We present the 0-shot re-ranking results with BRIDGE v2 using CodeBERT+GAT in Table B.3. Without any training data, CodeBERT+GAT improves BRIDGE v2's accuracy by 1.3%.

## C    Performance by Difficulty

In Table A.2, we break down the error detection performance in Table 2 by question difficulty and compare the performance of CodeBERT and CodeBERT+GAT. While CodeBERT can perform better on *easy* and *medium* questions for some parsers, CodeBERT+GAT consistently wins on *hard* and *extra hard* questions, showing the effectiveness of introducing structural features for harder questions. Since 63.1% of the questions are of *easy* or *medium* difficulty, the overall evaluation in Table 2 favors CodeBERT.

## D    Modified SQLite Grammar

we use a modified version based on a publicly available context-free grammar for SQLite https://github.com/antlr/grammars-v4/tree/master/sql/sqlite, We slightly modify the publicly available SQLite grammar for Antlr4[4] to introduce more hierarchical structures of SQL queries at the top level.

**Terminals**    We represent SQL keyword 'GROUP BY' by a single terminal GROUP_BY_ and 'ORDER BY' by ORDER_BY_. The original grammar reuses BY_ for 'BY', which we think breaks the entirety of these two keywords.

**Non-terminals**    We first remove values_clause and rules related to window functions, as they are not used by SQL queries in the Spider dataset. Then we break the select_core non-terminal, which represents a SQL query starting with SELECT, into a composition of multiple non-terminals, one for each SQL clause.

Our new SQLite grammar is listed as follows:

```
select_core:
  (
    SELECT_ (DISTINCT_ | ALL_)?
    result_clause
    (from_clause)?
    (where_clause)?
    (group_by_clause)?
  )
```

---

[3]We use all examples in examples/*_test.json in https://github.com/chiahsuan156/KaggleDBQA

[4]https://github.com/antlr/grammars-v4/tree/master/sql/sqlite

```
;

result_clause:
  result_column (COMMA result_column)*
;

from_clause:
  FROM_ table_or_subquery
  (COMMA table_or_subquery)*
  | FROM_ join_clause

;

where_clause:
  WHERE_ expr
;

group_by_clause:
  GROUP_ BY_ expr
  (COMMA expr)*
  (HAVING_ expr)?
;
```

The original grammar for `select_core`:

```
select_core:
  (
    SELECT_ (DISTINCT_ | ALL_)?
    result_column
    (COMMA result_column)*
    (FROM_
      (table_or_subquery
        (COMMA table_or_subquery)*
        | join_clause
      )
    )?
    (WHERE_ whereExpr=expr)?
    (GROUP_ BY_ groupByExpr+=expr
      (COMMA groupByExpr+=expr)*
      (HAVING_ havingExpr=expr)?
    )?
  )
;
```

Notice the excessive use of * in the original grammar that fails to represent the hierarchical relationship between the SELECT statement and each clause.

## E   Qualitative Beam Examples

As mentioned in Section 3.2, the three text-to-SQL parsers behave differently. We present their beam predictions on two samples in our training split in Table E.4. We can observe that SmBoP and NatSQL could generate more executable SQL queries than BRIDGE v2. Both SmBoP and NatSQL are capable of generating diverse errors, but NatSQL's beam predictions are more likely to share prefixes. As an example, SmBoP generates diverse SELECT clauses on both samples, while the SELECT clauses predicted by BRIDGE v2 and NatSQL do not change.

| Question: | How many heads of the departments are older than 56? |
|---|---|
| Gold SQL: | `SELECT COUNT(*) FROM head WHERE head.age > 56` |

| SmBoP | `SELECT COUNT(*) FROM head WHERE head.age > 56`
`SELECT head.name FROM head WHERE head.age > 56`
`SELECT MAX(head.age) FROM head WHERE head.age > 56`
`SELECT head.age FROM head WHERE head.age > 56`
`SELECT * FROM head WHERE head.age > 56` |
|---|---|
| BRIDGE v2 | `SELECT COUNT(*) FROM head WHERE head.age > 56` |
| NatSQL | `SELECT COUNT(*) FROM head WHERE head.age > 56`
`SELECT COUNT(*) FROM department WHERE department.department_id in (SELECT management.department_ID FROM management, head WHERE head.age = 56)`
`SELECT COUNT(*) FROM head WHERE head.age = 56`
`SELECT COUNT(*) FROM head WHERE head.age < 56`
`SELECT COUNT(*) FROM head WHERE head.age >= 56` |

| Question: | Show the names of the three most recent festivals. |
|---|---|
| Gold SQL: | `SELECT festival_detail.festival_name FROM festival_detail ORDER BY festival_detail.year DESC LIMIT 3` |

| SmBoP | `SELECT festival_detail.festival_name FROM festival_detail ORDER BY festival_detail.year DESC LIMIT 3`
`SELECT festival_detail.festival_name FROM festival_detail WHERE festival_detail.year = (SELECT MAX( festival_detail.year ) FROM festival_detail)`
`SELECT 3 FROM festival_detail WHERE festival_detail.year = (SELECT MAX( festival_detail.year ) FROM festival_detail)`
`SELECT MAX( festival_detail.year ) FROM festival_detail ORDER BY festival_detail.year DESC LIMIT 3`
`SELECT MAX( festival_detail.year ) FROM festival_detail ORDER BY festival_detail.year DESC` |
|---|---|
| BRIDGE v2 | `SELECT festival_detail.Festival_Name FROM festival_detail ORDER BY festival_detail.Year ASC LIMIT 3`
`SELECT festival_detail.Festival_Name FROM festival_detail ORDER BY festival_detail.Year LIMIT 3`
`SELECT festival_detail.Festival_Name FROM festival_detail ORDER BY festival_detail.Year DESC LIMIT 3` |
| NatSQL | `SELECT festival_detail.festival_name FROM festival_detail ORDER BY festival_detail.year DESC LIMIT 3`
`SELECT festival_detail.festival_name FROM festival_detail ORDER BY festival_detail.year ASC LIMIT 3`
`SELECT festival_detail.festival_name , festival_detail.year FROM festival_detail ORDER BY festival_detail.year DESC LIMIT 3`
`SELECT festival_detail.festival_name FROM festival_detail`
`SELECT festival_detail.festival_name FROM festival_detail GROUP BY festival_detail.festival_name ORDER BY festival_detail.year DESC LIMIT 3` |

Table E.4: Example beam predictions collected from three base parsers in our training dataset.