# OpenReview forum: "Error Detection for Text-to-SQL Semantic Parsing"
_EMNLP/2023/Conference — EMNLP 2023 Findings_

### Official Review · Reviewer_jWP3 · 2023-08-04

**Soundness:** 5

**Excitement:**

4: Strong: This paper deepens the understanding of some phenomenon or lowers the barriers to an existing research direction.

**Paper Topic And Main Contributions:**

The paper presents a semantic error detection approach for the output of text-to-SQL semantic parsers, consisting of (a) a method to generate a cross-domain error detection dataset, (b) a model for the error detection problem, and (c) a comprehensive study on error detection for text-to-SQL semantic parsing.
Specifically, the authors collect a dataset of errors using weak versions (trained on smaller datasets from fewer databases than their default version) of base parsers.
The proposed model uses an encoder (based on CodeBERT), whose input is the natural language questions and generated SQL queries. The output of the model is the aggregation of the output of the encoder and that of two graph models, one operating on natural language information (using dependency and constituency trees) and one operating on SQL information (using query abstract syntax trees). The model is fine-tuned with cross-entropy loss, where positive examples are gold (question, SQL) tuples, and negative examples come from the collected dataset of errors.
The authors report results on error detection, comparing their method with simple prediction probability thresholds and dropout-based uncertainty, showing that the proposed method outperforms the baselines. Moreover, the paper includes experiments on cross-parser generalisation, reranking of beam results, and action triggering in interactive systems, using three different parsers with different decoding implementations.
Interestingly, the proposed method is not tied to a specific parser implementation, and can be used for the output of a number of parsers.

**Questions For The Authors:**

* line 267: can you provide an example of the removed information? why is it that they “do not carry much semantic information"?
* have you considered training the proposed model on the output of more than one parser (e.g. BRIDGE+NatSQL) and test on the other (e.g. SmBoP)?

**Reasons To Accept:**

* Given the recent success of text-to-SQL parsers, the interest and importance of the task of error detection in the output of text-to-SQL systems is increasing in the community.
* The proposed method reports strong experimental results. Moreover, the paper includes extensive experimentation on a number of scenarios that are relevant to error detection and handling, using a number of parsers.
* The experiment described on section 4.2.4 (Action Triggering in Interactive Systems) is a strong point of the paper; it views the error detection problem from a more abstract focus point and considers the practical implications of parser errors.
* The paper is very well written and easy to follow.

**Reasons To Reject:**

n/a

**Reproducibility:**

4: Could mostly reproduce the results, but there may be some variation because of sample variance or minor variations in their interpretation of the protocol or method.

**Reviewer Confidence:**

4: Quite sure. I tried to check the important points carefully. It's unlikely, though conceivable, that I missed something that should affect my ratings.

**Typos Grammar Style And Presentation Improvements:**

* line 102: it might be a good idea to start with the phrase “Most existing neural text-to-SQL parsers” instead of “Existing text-to-SQL parsers”, since there is a number of (mainly older) approaches that do not follow any of the three decoding mechanisms mentioned.

---

> ### Author Rebuttal · Authors · 2023-08-29
>
> We appreciate the reviewer’s effort in reviewing our submission and are grateful for their recognition of our work.
>
> **[ln267: Example of the removed information and why]**
>
> We remove two kinds of information during preprocessing: foreign key relations  (i.e. ON tab1.col1 = tab2.col2) and related keywords (i.e. JOIN and AS) in FROM clauses. Since foreign key relations are specific to the underlying database, it may not correspond to the semantics of the natural language question. In addition, foreign key relations usually consist of a lot of words, especially when joining more than two tables in the FROM clause. This could make the model prone to overfitting, as shown by our ablation studies (Table 6). Therefore, we only keep the table names in the simplified SQL query.
>
> An example from the Spider dev set:
>
> Question:
>
> List all singer names in concerts in year 2014.
>
> Original SQL:
>
> ```sql
> SELECT T2.name FROM singer_in_concert AS T1 JOIN singer AS T2 ON T1.singer_id  =  T2.singer_id JOIN concert AS T3 ON T1.concert_id  =  T3.concert_id WHERE T3.year  =  2014
> ```
>
> Simplified SQL:
>
> ```sql
> SELECT singer.name FROM singer_in_concert singer concert WHERE concert.year  =  2014
> ```
>
> In this example, the original SQL has 26 words (segmented by whitespace) and the simplified SQL only has 10 words. Despite the massive difference in length, they convey the same semantics.
>
> **[Training the proposed model on the output of more than one parser]**
>
> We thank the reviewer for this suggestion and agree that mixing data from multiple parsers could potentially enhance the performance and generalization ability of error detection models. We did not include this setting in the paper to keep our experiment settings succinct.
>
> To address this question, we test the suggested setting by merging the training data collected from BRIDGE v2 and NatSQL. We train a CodeBERT + GAT model and test it on SmBoP, following the same cross-parser evaluation protocol as in Section 4.2.2.  The new results are presented in the last row of the following table (+ and - denotes positive and negative examples). The first 3 rows are taken from Table 3 in our paper.
>
> | Parser | Source |+Precision|+Recall|+F1|-Precision|-Recall|-F1|Acc|AUC|
> |:-|:-|:-:|:-:|:-:|:-:|:-:|:-:|:-:|:-:|
> SmBoP$^s$|/                      |81.5|91.9|85.7|**56.6**|25.3|29.4|76.9|79.2|
> CodeBERT + GAT|BRIDGE v2         |80.0|**95.9**|86.8|44.1|14.4|18.8|77.8|77.0|
> CodeBERT + GAT|NatSQL            |83.1|93.3|**87.6**|56.0|31.5|38.1|**79.9**|78.2|
> CodeBERT + GAT|BRIDGE v2 + NatSQL|**83.6**|92.4|87.4|55.5|**35.5**|**41.4**|79.8|**80.5**|
>
>
> Mixing training data generated by different parsers does enhance the coverage of errors and **improves the error detection model’s performance on negative test examples**, shown by the sizable improvement in negative recall and F1, **as well as overall performance** shown by the increase of AUC.
>
>
> **[Typos Grammar Style And Presentation Improvements]**
>
> We thank the reviewer for pointing out this issue. We will correct it in the revised version.

---

### Official Review · Reviewer_nSMo · 2023-08-05

**Soundness:** 3

**Excitement:**

3: Ambivalent: It has merits (e.g., it reports state-of-the-art results, the idea is nice), but there are key weaknesses (e.g., it describes incremental work), and it can significantly benefit from another round of revision. However, I won't object to accepting it if my co-reviewers champion it.

**Paper Topic And Main Contributions:**

This paper considers the task of error detection for text-to-SQL semantic parsing: given a prediction result, the model needs to decide whether the prediction is correct or not. This type of error detection models can be used for quality control (ensuring X% precision by filtering out bad predictions) and for reranking.

The error detection model is based on CodeBERT and graph attention networks. The graphs used are obtained through parse trees (dependency and constituency) on the utterance side, and abstract syntax trees on the SQL output side. Additionally, the paper proposes to train the error detection model with cross-domain data, which are generated by training the parsers on a subset of domains and running inference on a different subset of domains. (Here domains = databases in the Spider dataset).

Experiment results show that the proposed error detection model consistently outperforms using "confidence" measures directly from the three baseline parsers, SmBoP, BRIDGE v2, and NatSQL on error detection task. The paper also shows improved execution accuracy through reranking for BRIDGE v2 and NatSQL parsers.

**Questions For The Authors:**

How is the effect of beam size during data generation on the quality of the error detection model?

**Reasons To Accept:**

- The paper provides an interesting recipe for training error detection models. While the particular model architecture is not very novel, the additional of graph attention network is not critical to the performance, I find the cross-domain data generation method to be interesting and it proves to the most effective in the ablation studies.
- The paper shows promising results on error detection, simulated interactive semantic parsing, and reranking on a combination of base parser and experiment settings.

**Reasons To Reject:**

- The chosen base models are not strong enough. According the Spider leaderboard, the three models used in the paper are all below 15th rank. There are quite many options in the leaderboard with code available. It would be really informative to see whether the proposed model works for these stronger semantic parsers or not.  [This concern is addressed by the authors' responses below.]
- It is not clear how generalizable is the cross-domain data generation method to other datasets and settings. The Spider benchmark is set up specifically with a large variety of databases and the database split aims specifically at measuring cross-domain generalization. It is not unclear how whether the conclusion in the paper applies to in-domain evaluation as well or not.
- The reranking result with SmBoP is worse than not using reranking at all. This suggests the proposed model might not work so well with strong models (SmBoP is the strongest base model out of the three tested). Also see the first point in this section.

**Reproducibility:**

3: Could reproduce the results with some difficulty. The settings of parameters are underspecified or subjectively determined; the training/evaluation data are not widely available.

**Reviewer Confidence:**

3: Pretty sure, but there's a chance I missed something. Although I have a good feel for this area in general, I did not carefully check the paper's details, e.g., the math, experimental design, or novelty.

**Typos Grammar Style And Presentation Improvements:**

It is not clear how the dependency parse trees and constituency trees are merged (line 260).

The terms "hit" and "miss" are not explained in the paper, and it took me a while to figure out what they mean.

The terminology of "zero-shot" in section 4.2.2 is misleading. The error detection model is indeed trained on a different parser, but the training data and databases are seen if I understand correctly.

---

> ### Author Rebuttal · Authors · 2023-08-29
>
> We are grateful that the reviewer finds our cross-domain data generation method “interesting and effective” and our experiment results “promising.” We also thank the reviewer for the valuable constructive comments.
>
> **[The chosen base models are not strong enough]**
>
> We agree with the reviewer that our error detection model should work for  stronger parsers as well. In this work, we prioritized covering different decoder architectures and picked strong parsers at the time of our experiments. To address this concern, now we report new results on RESDSQL + NatSQL[1] , which is the strongest open-source parser and ranks 5th on the Spider leaderboard (as of 8/28/23). Due to time and computing restraints, we test the RESDSQL + T5-large + NatSQL variant using the released checkpoint [2], which is still stronger than SmBoP (81.9% vs 75.0% EX on Spider dev as reported by their papers). Since RESDSQL uses a sequence-to-sequence decoder, we compare against using its prediction probability as the uncertainty baseline. We follow the same cross-parser evaluation setup in Section 4.2.2 and test the CodeBERT + GAT model trained on data collected from NatSQL. The results are in the following table (+ and - denotes positive and negative examples):
>
> |Parser | +Precision | +Recall | +F1 |-Precision| -Recall | -F1 | Acc | AUC |
> |:-:|:-:|:-:|:-:|:-:|:-:|:-:|:-:|:-:|
> RESDSQL$^p$          |**82.8**|91.7|**87.0**|50.3|31.3|38.4|78.8|79.4|
> CodeBERT + GAT (ours)|82.7|**92.5**|**87.0**|**54.3**|**32.9**|**38.7**|**78.9**|**81.0**|
>
>
> Our model performs comparably or better than RESDSQL$^p$ on all metrics, which is consistent with our cross-parser generalization results (Table 3). The improvement is most salient under the AUC metric, which is our main metric. **Notably, this is generalizing from a weak parser to a stronger one.** We can reasonably expect our method to perform even better with data collected from RESDSQL itself, which we will do in the revised version.
>
> **[It is unclear whether the conclusion in the paper applies to in-domain evaluation as well]**
>
> We agree that error detection for in-domain settings is also an important aspect. In this work, we focus on the Spider dataset because it has been considered more challenging than text-to-SQL datasets with in-domain evaluation settings (e.g., single-database datasets such as Geoquery [4]), due to its cross-domain setup and more complicated SQL structures. In light of the promising performance of our approach on this more challenging dataset, we are optimistic about adapting our method to in-domain evaluation settings.
>
> **[The effect of beam size during data generation]**
>
> There are two factors we take into consideration when generating our training data. (1) It is important to collect diverse executable errors. This is shown by the statistics of generated data (Table 1) and results of cross-parser generalization (Sec. 4.2.2 and Table 3). (2) The collected data should have the same error distribution as the parser applied at test time. We use the same hyperparameter as they are in the official implementation of each parser during data collection
>
> Hence, changing beam size during beam search may have a mixed influence on data quality. Intuitively, using a larger beam size would help to generate more diverse data that has broader coverage of different types of errors. However, it could also deviate from the real error distribution in testing. As discussed in the Limitations section, future work could explore more principled data synthesis approaches.
>
> **[How the dependency parse trees and constituency trees are merged]**
>
> Since edges in a dependency parse tree are between two actual tokens in a text snippet, which are leaf nodes in a constituency tree, we directly add these edges to the constituency parse tree. We will add more detailed descriptions to clarify this in the revised version.
>
> **[The use of “hit”, “miss”, and “zero-shot” are confusing]**
>
> We apologize for the confusion caused by these terms. The use of “hit” and “miss” are borrowed from the notion of beam hit and beam miss in beam search used in previous work such as Bertrand-DR [3]. Your understanding of “zero-shot” is correct. We will remove the term “zero-shot” in Section 4.2.2. Thanks for pointing these out and we will improve the clarity of the writing in the revision.
>
> References:
>
> [1] Li, Haoyang, Jing Zhang, Cuiping Li, and Hong Chen. "Resdsql: Decoupling schema linking and skeleton parsing for text-to-sql." In Proceedings of the AAAI Conference on Artificial Intelligence, vol. 37, no. 11, pp. 13067-13075. 2023.
>
> [2] https://github.com/RUCKBReasoning/RESDSQL#step1-prepare-checkpoints
>
> [3] Kelkar, Amol, Rohan Relan, Vaishali Bhardwaj, Saurabh Vaichal, Chandra Khatri, and Peter Relan. "Bertrand-dr: Improving text-to-sql using a discriminative re-ranker." arXiv preprint arXiv:2002.00557 (2020).
>
> [4] Zelle, John M., and Raymond J. Mooney. "Learning to parse database queries using inductive logic programming." In Proceedings of the national conference on artificial intelligence, pp. 1050-1055. 1996.

---

### Official Review · Reviewer_12YA · 2023-08-09

**Soundness:** 3

**Excitement:**

2: Mediocre: This paper makes marginal contributions (vs non-contemporaneous work), so I would rather not see it in the conference.

**Paper Topic And Main Contributions:**

This paper proposes a parser-independent error detection model for text-to-SQL semantic parsing task. This error detection model utilizes the compositional structures in natural language and the corresponding SQL queries, and it outperforms other parser-dependent models and could generalize to unseen parsers.

Meanwhile, the author demonstrated that this error detection obtained gains in several tasks, such as error detection, SQL re-ranking and action triggering.

**Reasons To Accept:**

The error detection model proposed by this paper obtained good performance in multiple tasks, i.e., error detection, SQL re-ranking and action triggering. Meanwhile, this error detection model is independent with parsers.

**Reasons To Reject:**

1. The idea is not novel, this error detection model is just a semantic matching model between natural question and SQL query.
2.  This error detection model is independent with parsers, it need to be trained, and it need to be called during inference.

**Reproducibility:**

4: Could mostly reproduce the results, but there may be some variation because of sample variance or minor variations in their interpretation of the protocol or method.

**Reviewer Confidence:**

4: Quite sure. I tried to check the important points carefully. It's unlikely, though conceivable, that I missed something that should affect my ratings.

---

> ### Author Rebuttal · Authors · 2023-08-29
>
> We appreciate the time and effort that the reviewer put into reviewing our submission.
>
> **[The idea is not novel]**
>
> The contribution of this work is not limited to the design and implementation of the proposed error detection model. Besides building the **first parser-independent error detection model for text-to-SQL parsing**, which the reviewer finds to have "obtained good performance in multiple tasks,” our work is also the **first comprehensive study on error detection for text-to-SQL parsing**. This includes task formulation, evaluation settings, and performance comparison between several error detection methods.
>
> **[This error detection model is independent with parsers, it need to be trained, and it need to be called during inference]**
>
> Training a parser-independent error detection model like ours has a few advantages: (1) It can be applied to black-box semantic parsers (2) It can generalize well to unseen parsers. In our experiments, we compared against alternative methods that are parser-dependent and do not need to be trained, such as probability and drop-out based confidence measures, and showed that our proposed method can perform better.

---

### Meta-Review · Area_Chair_wfvM · 2023-09-16

**Recommendation:** 2

**Metareview:**

The authors propose a re-ranker model for text-to-SQL generations. The model is build on top of CodeBERT with extra GNN on top.

Reviewers have different views for this paper:
Reviewers 12YA and nSMo recognize the positive results presented in the paper, but state that such the concept of a re-ranker is well explored and that the baseline parsers on top of which the re-ranker is applied are not the best text-to-SQL models out there.
I tend to agree that it is not clear how this work is different than past work that presents a re-ranker -- the concept is not new, and was specifically proposed for semantic parsing in the past. I also believe the following reference is missing -- "Reranking for Neural Semantic Parsing". I would expect stronger baseline parsers and comparison to past re-rankers.

On the other hand, reviewer jWP3 recognized the positive results and the extensive experiments, and sees no issues with the paper.

---

### Decision · Program_Chairs · 2023-10-07

**Decision:**

Accept-Findings

**Comment:**

The authors propose a re-ranker model for text-to-SQL generations. The model is build on top of CodeBERT with extra GNN on top.

Reviewers have different views for this paper:
Reviewers 12YA and nSMo recognize the positive results presented in the paper, but state that such the concept of a re-ranker is well explored and that the baseline parsers on top of which the re-ranker is applied are not the best text-to-SQL models out there.
I tend to agree that it is not clear how this work is different than past work that presents a re-ranker -- the concept is not new, and was specifically proposed for semantic parsing in the past. I also believe the following reference is missing -- "Reranking for Neural Semantic Parsing". I would expect stronger baseline parsers and comparison to past re-rankers.

On the other hand, reviewer jWP3 recognized the positive results and the extensive experiments, and sees no issues with the paper.